# Phytochemical and Antioxidant Profile of the Medicinal Plant *Melia azedarach* Subjected to Water Deficit Conditions

**DOI:** 10.3390/ijms232113611

**Published:** 2022-11-06

**Authors:** Maria Celeste Dias, Diana C. G. A. Pinto, Maria Costa, Márcia Araújo, Conceição Santos, Artur M. S. Silva

**Affiliations:** 1Centre for Functional Ecology, Department of Life Sciences, University of Coimbra, Calçada Martim de Freitas, 3000-456 Coimbra, Portugal; 2LAQV/REQUIMTE, Department of Chemistry, Campus Universitário de Santiago, University of Aveiro, 3810-193 Aveiro, Portugal; 3IB2 Lab, Department of Biology & LAQV/REQUIMTE, Faculty of Sciences, Rua do Campo, Alegre, University of Porto, 4169-007 Porto, Portugal; 4Centre for the Research and Technology of Agro-Environmental and Biological Sciences (CITAB), University of Trás-os-Montes and Alto Douro, 5001-801 Vila Real, Portugal

**Keywords:** *Melia azedarach*, drought, water deficit, oleamide, catechols, flavonoids

## Abstract

Environmental stress triggered by climate change can alter the plant’s metabolite profile, which affects its physiology and performance. This is particularly important in medicinal species because their economic value depends on the richness of their phytocompounds. We aimed to characterize how water deficit modulated the medicinal species *Melia azedarach’s* lipophilic profile and antioxidant status. Young plants were exposed to water deficit for 20 days, and lipophilic metabolite profile and the antioxidant capacity were evaluated. Leaves of *M. azedarach* are rich in important fatty acids and oleamide. Water deficit increased the radical scavenging capacity, total phenol, flavonoids, and catechol pools, and the accumulation of β-sitosterol, myo-inositol, succinic acid, sucrose, d-glucose and derivatives, d-psicofuranose, d-(+)-fructofuranose, and the fatty acids stearic, α-linolenic, linoleic and palmitic acids. These responses are relevant to protecting the plant against climate change-related stress and also increase the nutritional and antioxidant quality of *M. azedarach* leaves.

## 1. Introduction

According to the Intergovernmental Panel on Climate Change [1], the increasing frequency of extreme weather events related to climate change, such as droughts and heatwaves, represents a severe threat to agriculture and forestry ecosystems. Moreover, the climatic projections forecast indicate that the climate change impacts on the functioning of the ecosystem will increase in intensity and frequency, largely due to the occurrence of more severe drought events [2]. Drought severely impacts plant development and productivity. The photosynthetic apparatus is one of the main targets of drought stress, decreasing net CO_2_ assimilation rate and PSII efficiency [3]. Plants respond to drought stress by triggering several defense mechanisms. The activation of the antioxidant system, which comprises enzymatic and non-enzymatic defense mechanisms, plays an important role in controlling reactive oxygen species (ROS) excessive production. The enzymatic system includes several enzymes, such as superoxide dismutase, catalase, ascorbate peroxidase, and glutathione peroxidase, and the non-enzymatic system comprises the ascorbate, glutathione, carotenoids, tocopherols, and phenolic compounds [4]. Within the phenolic compounds, the flavonoids are involved in plant physiological functions, demonstrating protective effects against biotic and abiotic stresses, including drought and UV-B radiation [5]. These compounds can scavenge the free radicals O_2_^•−^, OH^•^, and non-radicals such as ^1^O_2_, particularly the *o*-dihydroxy B-ring substituted flavonoids, such as the quercetin 3-*O*-glycosides and luteolin 7-*O*-glycosides, which exhibit a high antioxidant activity [6,7]. The coordinated action of both enzymatic and non-enzymatic defense systems contributes to controlling the levels of oxidative stress, therefore helping the plants cope with stress. In addition to these antioxidant responses, other protective metabolites from primary and secondary metabolic pathways can also be upregulated by abiotic stress-inducing changes in metabolite pools that can affect the pharmaceutical characteristics of plants [8,9,10,11]. For instance, drought conditions enhanced the levels of several secondary metabolites such as (E) β-caryophyllene in *Origanum vulgare*, rutin, quercetin, and betulinic acid in *Hypericum Brasiliense*; oleanolic acid and botulin in *Betula platyphylla*; glycyrrhizin in *Glycyrrhiza glabra*; cyanogenic glucosides in *Phaseolus vulgaris; p*-coumaric acid, rutin, luteolin and luteolin-7-*O*-glycoside, apigenin, and 1,3 dicaffeoylquinic acid in *Achillea pachycephal;* and lupeol derivative and ursolic acid in *Olea europaea* [11,12,13]. In *Artemisis annua* and *Glycine max*, drought conditions increased the contents of sterols, promoting tolerance to dehydration [14,15]. In olive plants, drought, or drought combined with heat and high UVB radiation, increased the levels of fatty acids (e.g., palmitic and oleic acids), helping to protect from membrane damage and promoting the accumulation of osmoprotectant agents (such as glucose and sorbitol) [7,13]. Moreover, the adjustments in the levels of flavonoids (e.g., luteolin and derivatives (luteolin-7-*O*-glucoside and luteolin-4-*O*-glucoside), apigenin-7-*O*-glucoside, chrysoeriol-7-*O*-glucoside, and kaempferol derivatives), secoiridoids (oleuropein and oleoside-11-methyl ester), and hydroxycinnamic acid derivatives (verbascoside) [7] in response to drought, together with the activation of peroxidases enzymes [13], played an important role in the control of oxidative stress. In Balangu plants (*Lallemantia sp.*), the levels of proline and total phenolic compounds, as well as the activity of antioxidant enzymes (e.g., superoxide dismutase (SOD) and ascorbate peroxidase (APX)), increased drought stress tolerance [16]. Additionally, in *Coleus plectranthus* and *Coleus forskholii*, drought increased the activity of antioxidant enzymes (e.g., SOD, APX, and catalase) along with the accumulation of non-enzymatic antioxidants (such as α-tocopherol, reduced glutathione and ascorbic acid) [11]. The production of sucrose, anthocyanins, total flavonoids, and phenolics, and the activity of the enzyme phenylalanine lyase (involved in the biosynthesis of flavonoids and phenylpropanoids) was enhanced in *Labisia pumila* in response to drought conditions [11]. These changes are particularly relevant in medicinal species since they are rich in important bioactive compounds that have high applicability in the pharmaceutical and food industries [16] as well as in the healthcare of most of the world’s population [17]. Medicinal species are a source of natural compounds with antioxidant properties, namely secondary metabolites, such as phenols, flavonoids, terpenes and terpenoids, carotenoids, alkaloids and saponins, and primary metabolites, such as amino acids, fatty acids, carbohydrates, and intermediates of the tricarboxylic acid (TCA) cycle [12].

*Melia azedarach* is considered a multipurpose tree native to West Asia. Still, it is nowadays cultivated in several countries (such as India, Pakistan, Nepal, Sri Lanka, East Timor, Indonesia, Lebanon, Palestine, Syria, Tunisia, Algeria, Cyprus, Greece, Argentina, China, Uganda, Kenya, Brazil, Australia, Southern France, Northern Italy, Croatia, and Portugal) with temperate to warm climates [18,19,20,21]. However, the projected increase in the frequency of extreme weather events in many regions of the globe during this century may reshape *M. azedarach* future suitable habitats [21]. This species has economic interest since it produces timber of high quality, which is often used to make furniture, farm tools, boats, vehicles, plywood, toys, and musical instruments and is largely used as an ornamental tree due to its ability to withstand a wide range of climatic and soil conditions [18,21,22]. Moreover, extracts of the different parts of this medicinal species are shown to have pharmacological and toxicological characteristics. For instance, leaves are rich in several bioactive molecules, such as flavonoids, terpenoids, limonoids, fatty acids, carbohydrates, steroids, alkaloids, saponins and tannins [23,24], which have been demonstrated to have antibacterial, antifungal, antiparasitic, antiviral and insecticide properties [19,25,26].

The global recognition of the health benefits of plant phytocompounds promotes an increased use in the pharmaceutical and food industries and a search for novel natural antioxidant compounds [16]. This tendency highlights the need to tackle how climate change-related stressors modulate medicinal species’ phytochemical composition and redox state [7,8,12,16]. Thus, we aim to study how *M. azedarach* plants adjust their metabolite and antioxidant status to deal with a water deficit condition and decipher how drought modulates leaves’ bioactive compounds, promoting their nutritional and economic value. We hypothesized that *M. azedarach* metabolite plasticity to drought conditions involves an adjustment of the lipophilic metabolites and antioxidant status.

## 2. Results and Discussion

Climate change stressors can reduce plants’ growth and phytocompound levels [14]. We reported previously that water deficit (WD) conditions reduce *M. azedarach* water potential, photosynthesis, and cell membrane stability but activate the antioxidant enzyme system (e.g., superoxide dismutase, ascorbate peroxidase, glutathione reductase, and catalase) and increase the ascorbate pool [22].

A commonly used marker to evaluate the level of plant dehydration or water status is the leaf relative water content (RWC). In the present work, *M. azedarach* plants under well-watered (WW) conditions presented a leaf RWC of 91.1 ± 1.77%, while the ones under WD conditions showed a leaf RWC of 74.1 ± 4.19%. These data indicated that WD treatment induced severe stress since the RWC levels decreased to values below 80% [27]. Severe drought stress can cause several physiological impairments in plants, such as a decrease in photosynthesis, and when this stress condition is extended for a long period, plant growth and productivity can be compromised [28].

In the present work, we demonstrated for the first time that drought conditions significantly increase the antioxidant status of this species (increase in total antioxidant capacity, total phenols, catechols, and flavonoid pools; see Figure 1). Similar results were obtained in other species such as *Triticum aestivum*, *H. brasiliens*, *L. pumila*, *C. forskholii* and *C. amboinicus*, *O. europaea*, and Achillea sp. (*A. pachycephala*, *A. millefolium*, *A. nobilis*, and *A. filipendulina*) [11,29,30]. Drought seems to regulate the biosynthesis of phenolics, increasing the activities of key enzymes, such as phenylalanine lyase (PAL) and chalcone synthase (CHS). Moreover, drought upregulates the transcript levels of genes encoding key biosynthetic enzymes of the phenylpropanoid and flavonoid biosynthesis, such as PAL, CHS, chalcone isomerase, dihydroflavonol 4-reductase, flavonol synthase, flavanone-3-hydroxylase, cinnamate 4-hydroxylase, 4-coumarate-CoA ligase, dihydroflavonol-4-reductase, anthocyanidin synthase, and UDP flavonoid glycosyltransferase [30,31].

The GC-MS analysis allowed the identification of six classes of compounds in the hexane extracts, long-chain alkanes (tetracontane (C_40_H_82_), hexatriacontane (C_36_H_74_) and dopentacontane (C_52_H_106_)), sterols (β-sitosterol, stigmasterol, and campesterol), the polyalcohol myo-inositol, the amide oleamide, organic acids (linoleic, α-linolenic, palmitic, stearic and succinic acids), and carbohydrates (sucrose, melibiose, d-glucose, d-(+)-talofuranose, d-psicofuranose, and d-(-)-fructofuranose) (Table 1). The organic acids, oleamide, and long-chain alkanes were the most abundant compounds found in the hexane extract (Table 1). Some of these compounds (e.g., sterols such as β-sitosterol, long-chain alkanes, and fatty acids such as the palmitic, stearic, and linoleic acids) were reported previously in *M. azedarach* leaves [32,33]. The fold change in metabolites (Table 1) after WD treatment showed that only the tetracontane, stigmasterol, campesterol, oleamide, linoleic acid, and d-(+)-talofuranose exhibit a negative response (decrease in the respective metabolite).

Long-chain alkanes are presented in high amounts in *M. azedarach* leaves, and WD conditions stimulated the accumulation of these compounds (Table 1). Leaf cuticular wax is mainly composed of long-chain alkanes, triterpenes, alcohols, aldehydes, and fatty acids [34]. The increase in long-chain alkanes in WD *M. azedarach* plants may represent an investment to strengthen cuticle structure and, therefore, a protective barrier against water loss.

Within the organic acids, *M. azedarach* leaves are particularly rich in the polyunsaturated fatty acids α-linolenic and linoleic acids, which are compounds that decrease the risk of human cardiovascular disease [35], and also in the saturated fatty acids, palmitic and stearic acids. Moreover, WD promoted the increase in the levels of these fatty acids (except linoleic acid) (Table 1). Studies in *Populus simonii* plants suggested that drought stress promotes the biosynthesis of long-chain fatty acids (LCFAs) through the upregulation of the enzyme acetyl-CoA carboxylase 1 involved in the elongation of LCFAs [36]. However, the mechanisms behind the fatty acid adjustments in response to drought are not well understood.

The TCA-cycle intermediate, succinic acid, was also stimulated by the WD treatment and is positively correlated with the antioxidant battery, ABTS, total phenols, catechols, and flavonoids (Table 1, Figure 2; r ≥ 0.85 and *p* ≤ 0.04). In fact, this acid (and its derivatives) has antioxidant properties and plays an important role as an antidiabetic agent for type 2 diabetes, lowering blood glucose and glycosylated hemoglobin [37]. In addition, several studies also demonstrated that the treatment with this acid improves motor behavior and ameliorates the cognitive deficits in animal models with neurodegenerative diseases [38].

To our knowledge, this is the first report of oleamide presence in *M. azedarach* leaves. The levels of oleamide were not affected by the WD treatments, despite some works referred this compound putative stress protector [8]. So, the role of oleamide in plant physiology remains unknown, possibly acting in growth and development regulation and pathogen interactions [39]. Moreover, oleamide is an emerging drug with several pharmacological properties (e.g., neuro-signaling and antitumoral) [40], and the high content of this amide (compared to other species, such as *Coxi lachrymal-jobi* [41], *Lemna minor* [42], *Nigella sativa* [43] and *O. europaea* [8]) make *M. azedarach* a valuable source of this compound.

The leaves of *M. azedarach* are also rich in sterols, and we identified sitosterol, stigmasterol, and campesterol (Table 1). Along with sphingolipids and glycerolipids, sterols are structural components of cell membranes that maintain membrane permeability and fluidity [15]. Under abiotic stress conditions, changes in sterols’ total content, as well as variations in their profile (particularly the composition ratio of sitosterol and stigmasterol) can occur affecting the state of the cell membrane [29,44,45]. In the present work, only sitosterol increased in response to WD treatment, indicating that this treatment induced changes in the ratio of campesterol/sitosterol and sitosterol/stigmasterol, a process of stress compensation [44]. Nevertheless, considering that plant sterols reduce human total and LDL cholesterol by inhibiting cholesterol absorption [46], the increase in sitosterol levels by the WD treatment represents an improvement of the nutritional quality of *M. azedarach* leaves.

In *M. azedarach*, the levels of carbohydrates changed in response to WD, leading to a significant increase in sucrose, d-glucose and derivatives, d-psicofuranose, and d-(-)-fructofuranose (Table 1). The increase in the sugars pools is a typical response of plants to drought stress [3,13]. For instance, sucrose and glucose increase under water deficit conditions can act as osmolytes to maintain cell turgor, while the increase in other sugars, such as fructose, can be linked with the upregulation of some secondary metabolite synthesis [47]. Moreover, the increase in these soluble sugars in *M. azedarach* is positively correlated with total phenols, catechols, flavonoids, and total antioxidant activity (TAA) (evaluated by the ABTS assay) (except for the case of d-psicofuranose) (Figure 2; r ≥ 0.83 and *p* ≤ 0.04), suggesting their involvement in antioxidant responses, possibly helping in ROS balance and oxidative stress control, membrane protection, and enzyme/protein stabilization as reported by You and Chan [48]. Despite this role in stress tolerance, the increase in carbohydrate contents also represents an augmenting of energy availability, enriching the nutritional level of *M. azedarach* plants.

The polyalcohol myo-inositol is a plant signaling molecule and under drought conditions can act in osmoregulation processes and/or in oxidative control (e.g., ROS scavenger) [49]. This protective role seems to be important in *M. azedarach* plants, as their levels increase under WD conditions and a positive correlation with the antioxidant battery, TAA, total phenols, flavonoids, and catechols was observed (Table 1, Figure 2; r ≥ 0.85 and *p* ≤ 0.03). Additionally, in humans, myo-inositol and its phosphate derivatives exert many valuable positive effects, acting as an antidiabetic, anti-inflammatory, antioxidant, and anticancer agent [50]. Our work demonstrates that *M. azedarach* leaves can be enriched by myo-inositol through exposure to WD, potentially increasing the health benefits.

## 3. Materials and Methods

### 3.1. Plant Material and Stress Treatment

Seeds of *Melia azedarach* Linn. were germinated in plastic pots (500 mL) with turf and vermiculite (2:1). After germination, seedlings were grown in a climatic chamber at 20 ± 2 °C, a 16/8 h (day/night) photoperiod, and with a photosynthetic active radiation of 500 ± 20 μmol m^−2^ s^−1^ (250 ± 20 µmol m^−2^ s^−1^ at the top of the plants) provided by Osram lamps. Light intensity in the climate chamber was measured with the external PAR sensor of the LI-6400XT (Portable photosynthesis system, LI-COR Bioscience, Lincon, NE, USA). Before the beginning of the water deficit treatment (WD), all plants were well-watered (100% field capacity). Two-month-old plants (60 days-old) with a height of 13.5 ± 3.2 cm were randomly assigned as well-watered (WW) (plants watered at 100% of field capacity (*n* = 12)), and WD plants were watered at 20% of field capacity for 20 days (*n* = 12). The pots were watered at 100% or 20% field capacity by restoring the quantity of water lost every second day (water lost was measured by weighing the pots with a scale). After these treatments’ leaves from the third top node (the youngest fully expanded) were collected for determination of the relative water content. Additionally, leaf samples (collected above the second node from the bottom) were immediately frozen in liquid nitrogen and stored at −80 °C.

### 3.2. Plant Water Status

Plant water status was determined by measuring the leaf relative water content (RWC). The leaves’ fresh weight was determined, and then the leaves were immersed in distilled water for 24 h at 4 °C. After this period, the turgid weight was determined, and the leaves were dried to calculate the dry weight. The percentage of RWC was determined using the formula: (FW-DW)/(TW-DW) × 100, where the FW was the leaf fresh weight, the DW the leaf dry weight, and the TW the leaf turgid weight.

### 3.3. Total Antioxidant Activity, Total Polyphenols, Catechols, and Flavonoids

Approximately 100 mg of frozen leaf samples were ground with 1 mL of methanol [8] for the determination of the total antioxidant activity, total polyphenols, catechols, and flavonoids. After incubation for 30 min at 40 °C the homogenate was centrifuged (5000× *g* for 5 min at 4 °C) and the supernatants were used for analysis.

For the total antioxidant activity (TAA), determined by the ABTS^+•^ free cation radical scavenging activity method, one aliquot of the leaf extract (10 µL) was incubated for 10 min at 30 °C with 200 µL of ABTS (2,20-azinobis(3-ethylbenzothiazoline-6-sulphonic acid)) as described by Re et al. [51]. Then, the absorbance of the supernatant was recorded at 734 nm, and the total antioxidant activity was calculated using a calibration curve for the gallic acid. Total antioxidant activity is expressed as µM Gallic Acid Equivalents per mg of dry extract.

For the determination of total polyphenols, the Folin–Ciocalteu method was used. One aliquot of the leaf extract (20 µL) was homogenated with 405 µL of a Folin–Ciocalteu reagent, and 75 µL Na_2_CO_3_ (20%). After an incubation period of 30 min (at 37 °C) the supernatant was read at 765 nm. Total polyphenols content was determined based on a gallic acid calibration curve. Total polyphenols are expressed as µM Gallic Acid Equivalent per mg of dry extract.

Catechols were determined using the molybdate assay. A sodium molybdate solution (5%) was prepared in 50% methanol, and 40 µL were mixed with 160 µL of leaf extract [52]. After incubation at 20 °C for 15 min, the absorbance of the supernatant was recorded at 370 nm. Catechol contents were determined based on a gallic acid calibration curve. Catechols are expressed as µM Gallic Acid Equivalents per mg of dry extract.

For flavonoid determination, one aliquot of the extract (37.5 µL) was homogenized with methanol, 75 µL of NaNO_2_ at 5% and 75 µL of AlCl_3_ at 10%. After 6 min in dark, 125 µL of 1 M NaOH was added, and the absorbance was recorded at 510 nm. Flavonoids content was determined based on a rutin calibration curve. Total flavonoids are expressed as µM Rutin Equivalents per mg of dry extract.

### 3.4. Extraction of Metabolites and Chromatography Analysis

For each treatment (WW and WD) leaves of *M. azedarach* were ground in a mill. Leaf powder (10 g) was mixed with 100 mL of hexane. The leaf metabolites were extracted at room temperature in three cycles of magnetic stirring for 72 h. The hexane was evaporated using a rotatory evaporator and the extract solutions were left to dry. Before analysis samples were silylated. In a glass tube, 250 μL of pyridine, 250 μL of *N*,O-bis(trimethylsilyl)trifluoroacetamide, 50 μL of trimethylsilyl chloride, 450 μL of the leaf extract (~10 mg mL^−1^), and 450 μL of tetracosane (1.4 mmol L^−1^) were mixed and placed at 70 °C (in a water bath) for 35 min. After that, the silylated extracts were injected into the gas chromatography mass spectrometer (QP2010 Ultra Shimadzu, DB-5-J&W capillary column of 30 m × 0.25 mm id and a film thickness of 0.25 μm) as described by Dias et al. [7,8]. Briefly, helium was used as carrier gas (35 cm s^−1^) and the initial temperature was fixed to 80 °C for 5 min with a temperature rate of 4 °C min^−1^ up to 285 °C for 10 min. The injector worked at a temperature of 250 °C and transfer-line temperature was 290 °C with a split ratio of 1:50. The mass spectrometer functioned in the electron impact mode with energy of 70 eV, and data were stored at a rate of 1 scan s^−1^ (*m*/*z* 33–750). The temperature of the ion source was 250 °C. The peak obtained in the chromatograms were identified by comparison with the library entries of mass spectra database (WILEY Registry TM of Mass Spectra Data and NIST14 Mass spectral). For metabolite quantification, calibration curves of pure compounds representative of the chemical families identified in these extracts (palmitic acid, cholesterol, maltose, octadecanol, and tetradecane) were prepared and injected in the GC-MS as described above for the leaf extracts.

### 3.5. Statistical Analysis

Data were analyzed by *t*-test at a significance level set to 0.05 using the statistic program SigmaStat for Windows version 3.1 (Systat Software, San Jose, CA, USA). Pearson correlations were performed in the SigmaStat for Windows version 3.1 (Systat Software, San Jose, CA, USA). Fold change was determined in Microsoft^®^ Excel for windows (version 10).

## 4. Conclusions

We characterized the antioxidant status (total antioxidant activity, catechols, total phenols, and flavonoids) and the leaf lipophilic profile of the multipurpose species *M. azedarach* and demonstrated that leaves are a source of important fatty acids, polyalcohol, sterols, TCA cycle-related metabolites, amides, and carbohydrates. Moreover, we demonstrated for the first time in this species that metabolite adjustments and the increase in the total antioxidant pool (total antioxidant activity, polyphenols, flavonoids, and catechols) represent a protective response to drought, helping *M. azedarach* plants to survive and tolerate this stress. Drought conditions can stimulate the accumulation of important compounds, contributing to increase the nutritional and bioactive richness of this species and thus its commercial value.

## Figures and Tables

**Figure 1 ijms-23-13611-f001:**
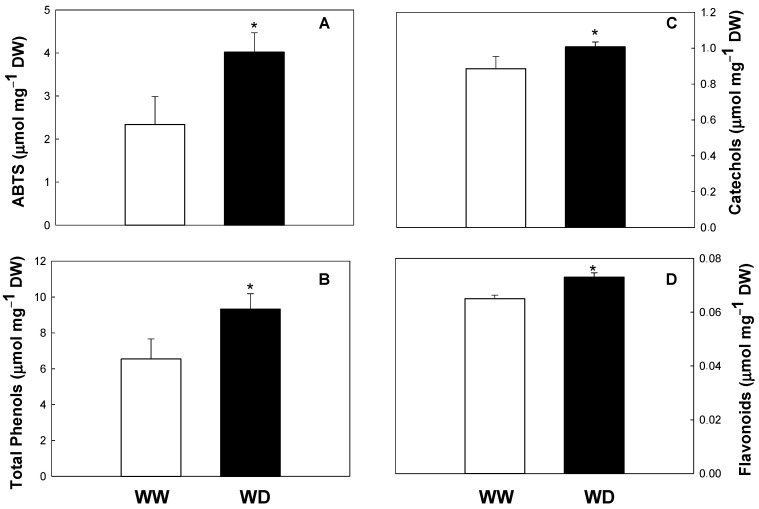
Total antioxidant activity evaluated by the ABTS assay (**A**), total phenolic content (**B**), catechols (**C**), and flavonoids (**D**) in well-watered (WW) and water-deficit (WD) plants of *Melia azedarach*. Values are means ± standard deviation (*n* = 6). Asterisks indicate statistical differences between treatments (*p* < 0.05).

**Figure 2 ijms-23-13611-f002:**
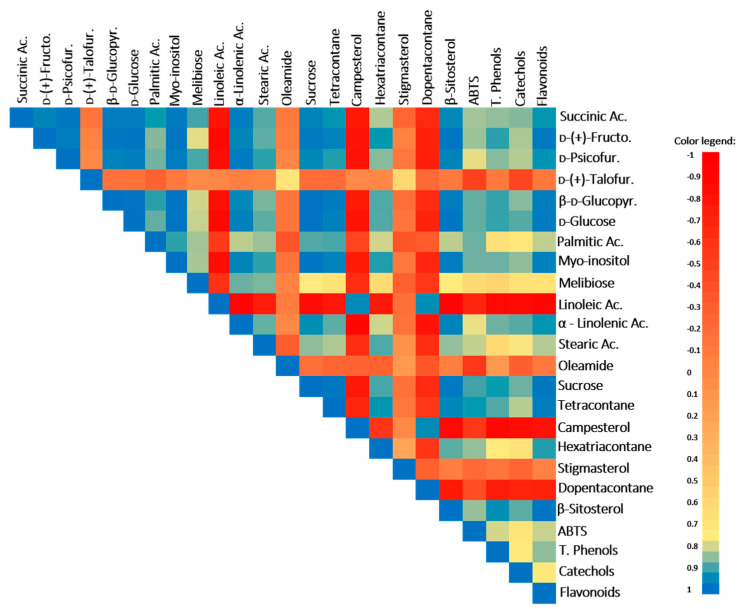
Pearson’s correlation coefficients between parameters in plants of the well-watered (WW) and water deficit (WD) conditions. Coefficients closer to 1 (blue color) mean a positive correlation, 0 (salmon color) indicates no correlation, and −1 (red color) means a negative correlation. Ac., acid; d-(+)-Fructo., d-(+)-Fructofuranose; d-(+)-Talofur., d-(+)-Talofuranose; d-Psicofur., d-Psicofuranose; β-d-Glucopyr., β-d-Glucopyranose; and LC, long chain.

**Table 1 ijms-23-13611-t001:** *Melia azedarach* leaf metabolite content (mg g^−1^ DW) in plants under in well-watered (WW) and water-deficit (WD) conditions. Values are means ± standard deviation (*n* = 3). For each line, asterisks indicate statistical differences between treatments (*p* < 0.05). Fold change (log_2_ (WD/WW)) in metabolites after WD treatment (values of the fold change are presented near the respective bar). Rt-retention time, DW-dry weight.

Rt (min)	Compound	Treatments (mg g^−1^ DW)		*p*-Value	Fold Change
		WW	WD		
	Long-chain alkanes				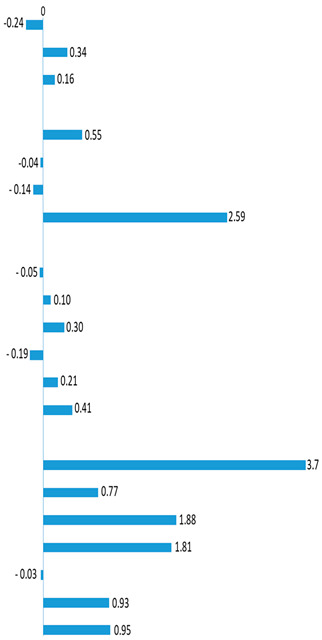
56.33	Tetracontane	2.10 ± 0.018	2.35 ± 0.039 *	<0.001	
61.12	Hexatriacontane	2.74 ± 0.105	3.48 ± 0.315 *	0.019	
66.16	Dopentacontane	3.28 ± 0.513	2.77 ± 0.066	0.165	
Sterols and polyalcohol					
64.43	β-Sitosterol	1.32 ± 0.097	1.94 ± 0.021 *	<0.001	
64.90	Stigmasterol	1.27 ± 0.110	1.24 ± 0.121	0.727	
60.44	Campesterol	1.41 ± 0.086	1.28 ± 0.036	0.072	
39.16	Myo-inositol	0.10 ± 0.02	0.58 ± 0.041 *	<0.001	
	Amides and organic acids				
45.57	Oleamide	3.42 ± 0.499	3.30 ± 0.345	0.758	
42.87	Stearic acid	2.81 ± 0.017	3.02 ± 0.105 *	0.027	
42.26	α-Linolenic acid	2.91 ± 0.131	3.59 ± 0.128 *	0.003	
41.52	Linoleic acid	3.38 ± 0.236	2.73 ± 0.034 *	0.040	
38.46	Palmitic acid	3.45 ± 0.190	3.99 ± 0.104 *	0.013	
22.25	Succinic acid	2.79 ± 0.005	3.72 ± 0.167 *	<0.001	
	Carbohydrates				
50.52	Sucrose	0.348 ± 0.055	4.54 ± 0.190 *	<0.001	
41.85	Melibiose	0.324 ± 0.006	0.555 ± 0.153	0.059	
36.90	d-Glucose	0.390 ± 0.072	1.43 ± 0.034 *	<0.001	
34.50	β-d-Glucopyranose	0.331 ± 0.029	1.16 ± 0.008 *	<0.001	
33.46	d-(+)-Talofuranose	0.386 ± 0.037	0.378 ± 0.012	0.729	
32.57	d-Psicofuranose	0.357 ± 0.044	0.680 ± 0.062 *	0.002	
32.35	d-(+)-Fructofuranose	0.361 ± 0.046	0.690 ± 0.002 *	<0.001	

## Data Availability

Data availability on request.

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
