# Peer review of "Phytochemical and Antioxidant Profile of the Medicinal Plant Melia azedarach Subjected to Water Deficit Conditions"

_ijms, 2022, doi:10.3390/ijms232113611_

Round 1

Reviewer 1 Report

The manuscript presents interesting results on the water deficit tolerance mechanism of Melia azedarach. 

Author Response

We would like to acknowledge the valuable comments, remarks and suggestions made by the Reviewers that helped us to improve the quality of the manuscript. We marked in yellow colour the changes made in the manuscript suggested by the Reviewers. 

Reviewer 1

Comment: The manuscript presents interesting results on the water deficit tolerance mechanism of Melia azedarach. 

Response: The authors acknowledge the Reviewer comment.

Reviewer 2 Report

The manuscript: "Phytochemical and Antioxidant Profile of the Medicinal Plant Melia azedarach Subjected to Water Deficit Conditions" represents an original study that evaluate the effect of water deficit in Melia azedarach on lipophilic profile and antioxidant status.     

I recommend it to be published in the International Journal of Molecular Sciences after major revision of the manuscript.

The following changes are recommended and some clarifications should be made:

Modification in the section Introduction

It is necessary to present in the Introduction section some previous results for the effect of drought stress on different plant species with respect to the secondary metabolite production, lipophilic profile or antioxidant state.

Pg. 2, Line 60-61: Please, add “cycle” after (TCA).

Modification in the section Results and Discussion

Pg. 2, Line 82: Please, define the abbreviation “WD” since it was mentioned here for the first time.

Pg. 2, Line 84-85: Also for the abbreviations “SOD, APX, GR…WW, RWC”.

Pg 2, Line 84-86: You should replace “In the present work” with “In that work…”.

Fig. 1: There are not symbols A,B,C,D in the Figure 1 for different parameters, as it is done in the caption. In the Figure 1 caption, the catechols should be symbolized with C and flavonoids with D.

Pg. 2: Please, compare your results for phenolics, flavonoids, catechols and antioxidant activity with other studies. What are the possible mechanisms for inducing the secondary metabolism in plants during the water deficit?

Pg. 4, Line 115-116: Is there any explanation for the enhancement levels of fatty acids in WD plants?

Pg. 4, Line 145-146: Please rephrase.

Pg 4, Line 152, 161: Please, define the abbreviation TAA. What is the differences between TAA and antioxidant battery, is it the same parameter for antioxidant activity measured by ABTS?

Pg 4, Line 160-162: Please add “plants” after “M. azedarach”.

Pg 4: Line 160-162: Are there any explanation for the positive correlations between myo-inositol and antioxidant activity and phenolics?

Pg 5. Table 1 caption: Please avoid starting the sentence with abbreviation.

Pg 5. Table 1: The content of the compounds (mg·mg-1 DW) should be inserted in the column with the treatments where the values of the parameters are included. Please, check the measurement unit mg·mg-1 DW. Define the abbreviation DW in the Table 1.

Pg 6. Figure 2: Please, define the abbreviations WW and WD in Figure 2 caption.

Pg 6. Figure 2: The Pearson’s correlation plot is confusing, since there are some color fields in the plot with different gradation of the blue color, ant this gradation in the blue color is not included in the scale that is positioned on the left side of the matrix. For example, it is difficult to note what is the correlation coefficient of sucrose with ABTS, catechols, or total phenols. It seems that color scale is not corresponding to the correlation plot with respect to coloration.

Modification in the section Material and Methods

Pg 6, Line 183-186: How the authors performed the experiments with field capacity for water?

Pg 7, Line 201-219: At the very last paragraph for each method (antioxidant activity, polyphenolics, flavonoids, catechols), it is necessary to include the measurement units for expression of the results. As example for ABTS = µM gallic acid equivalents per milligrams dry weight (µM GAE·mg DW).

Pg 8, Line 243-246: The subdivision “Statistical analysis” should be rephrased. What type of program was used for obtainment of Pearson correlation plot?

Modification in the section Conclusion

Pg 8, Line 248-250: The presence of phenols, flavonoids and catechols for this plant species is not mentioned in the conclusion.  

Pg 8, Line 250-254: The last sentence of the conclusion is not clear, it should be rephrased.

Author Response

We would like to acknowledge the valuable comments, remarks and suggestions made by the Reviewers that helped us to improve the quality of the manuscript. We marked in yellow colour the changes made in the manuscript suggested by the Reviewers. In what follows we describe the changes on the manuscript and we answer the Reviewers’ comments.

Reviewer 2

The manuscript: "Phytochemical and Antioxidant Profile of the Medicinal Plant Melia azedarach Subjected to Water Deficit Conditions" represents an original study that evaluate the effect of water deficit in Melia azedarach on lipophilic profile and antioxidant status. I recommend it to be published in the International Journal of Molecular Sciences after major revision of the manuscript. The following changes are recommended and some clarifications should be made:

Modification in the section Introduction 

Comment: It is necessary to present in the Introduction section some previous results for the effect of drought stress on different plant species with respect to the secondary metabolite production, lipophilic profile or antioxidant state.

Response: We added more information related to metabolites (phenolic and lipophilic compounds) and antioxidant enzyme response to drought in several species. Please see pg. 2 of the new version of the manuscript.

Comment: Pg. 2, Line 60-61: Please, add “cycle” after (TCA).

Response: We followed the Reviewer comment. Please see pg. 2.

Modification in the section Results and Discussion

Comments: Pg. 2, Line 82: Please, define the abbreviation “WD” since it was mentioned here for the first time. Pg. 2, Line 84-85: Also for the abbreviations “SOD, APX, GR…WW, RWC”.

Response: We followed the suggestion of the Reviewer, please see the new version of the manuscript (pg. 3).

Comment: Pg 2, Line 84-86: You should replace “In the present work” with “In that work…”.

Response: In fact, the data presented for the RWC was performed in the present work. We only presented these data in the text and do not provide a table or figure, because are only to averages (RWC in WW – well watered plants and WD - water deficit plants). We rephrased this paragraph in order to be clear that these data were collected in this work (please see pg. 3).

Comment: Fig. 1: There are not symbols A,B,C,D in the Figure 1 for different parameters, as it is done in the caption. In the Figure 1 caption, the catechols should be symbolized with C and flavonoids with D.

Response: We apologize for the missing of the symbols in the figures. We added this information in the new version of the figure 1. Please see pg. 4.

Comment: Pg. 2: Please, compare your results for phenolics, flavonoids, catechols and antioxidant activity with other studies. What are the possible mechanisms for inducing the secondary metabolism in plants during the water deficit?

Response: We followed the suggestion of the Reviewer, and we added new information concerning the results for other species and possible molecular mechanisms involved on secondary metabolism stimulation by drought (pg. 3).

Comment: Pg. 4, Line 115-116: Is there any explanation for the enhancement levels of fatty acids in WD plants?

Response: We provided a putative explanation for fatty acids increase, despite the low information available on this issue (please see pg. 5). 

Comment: Pg. 4, Line 145-146: Please rephrase.

Response: We rephrase the sentence. Please see pg. 5 “In M. azedarach, the levels of carbohydrates changed in response to WD, leading to an increase of sucrose, D-glucose and derivatives, D-psicofuranose, and D-(-)-fructofuranose significantly (Table 1).

Comment: Pg 4, Line 152, 161: Please, define the abbreviation TAA. What is the differences between TAA and antioxidant battery, is it the same parameter for antioxidant activity measured by ABTS?

Response: We corrected this sentence. TAA represent the total antioxidant activity and was evaluated by the ABTS assay. Please see pg. 5.

Comment: Pg 4, Line 160-162: Please add “plants” after “M. azedarach”.

Response: We followed the Reviewer comment. Please see pg. 5.

Comment: Pg 4: Line 160-162: Are there any explanation for the positive correlations between myo-inositol and antioxidant activity and phenolics?

Response: We were not able to find an explanation, or a relation of myo-inositol with these antioxidant parameters. We know that myo-inositol is a plant signaling molecule and under drought conditions can help, for example in the control of oxidative stress. Similarly, the phenolics, flavonoids or catechols also have an antioxidant role, helping in the control of oxidative stress.

Comment: Pg 5. Table 1 caption: Please avoid starting the sentence with abbreviation.

Response: We followed the Reviewers comment. Please see pg. 6, Table 1.

Comment: Pg 5. Table 1: The content of the compounds (mg·mg-1 DW) should be inserted in the column with the treatments where the values of the parameters are included. Please, check the measurement unit mg·mg-1 DW. Define the abbreviation DW in the Table 1.

Response: We inserted the units of the compounds in the column of the treatment, defined the abbreviature DW and also checked the units (mg/g DW). Please see pg. 6, table 1.

Comment: Pg 6. Figure 2: Please, define the abbreviations WW and WD in Figure 2 caption.

Response: We defined the abbreviations in the figure caption. Please see pg. 7.

Comment: Pg 6. Figure 2: The Pearson’s correlation plot is confusing, since there are some color fields in the plot with different gradation of the blue color, ant this gradation in the blue color is not included in the scale that is positioned on the left side of the matrix. For example, it is difficult to note what is the correlation coefficient of sucrose with ABTS, catechols, or total phenols. It seems that color scale is not corresponding to the correlation plot with respect to coloration.

Response: We understand the Reviewer comment. We rearranged the figure and the color scale. Moreover, we also introduced in the text the Pearson correlation coefficient and the P value every time that we mention the correlations (pgs. 5 and 6). Please see the new version of the figure 2, pg. 7.

Modification in the section Material and Methods

Comment: Pg 6, Line 183-186: How the authors performed the experiments with field capacity for water?

Response: The field water capacity is the amount of water held in soil after the excess water drained away. To measure this amount/quantity of water that is retained in the soil we used a scale. We previously weighed the empty pots, the soil in the respective pot, as well as the water needed to saturate the soil (100%FC) in each pot.

Plants were weighted every second day, and the water was replaced until the 100% or 20% (according to the respective treatment). We clear this point in the manuscript “The pots were watered at 100% or 20% field capacity by restoring the quantity of water lost every second day (water lost was measured by weighing the pots with a scale)”. Please see pg. 8.

Comment: Pg 7, Line 201-219: At the very last paragraph for each method (antioxidant activity, polyphenolics, flavonoids, catechols), it is necessary to include the measurement units for expression of the results. As example for ABTS = µM gallic acid equivalents per milligrams dry weight (µM GAE·mg DW).

Response: We added this information in M&M. Please see pg. 8 section 3.3.

Comment: Pg 8, Line 243-246: The subdivision “Statistical analysis” should be rephrased. What type of program was used for obtainment of Pearson correlation plot?

Response: We rephrased this sentence in the statistical analysis section. Please pg. 9.

Modification in the section Conclusion

Comment: Pg 8, Line 248-250: The presence of phenols, flavonoids and catechols for this plant species is not mentioned in the conclusion. Pg 8, Line 250-254: The last sentence of the conclusion is not clear, it should be rephrased.

Response: We followed the Reviewer comment, and we rephrased these sentences. Please see pg. 9.

Reviewer 3 Report

The submitted manuscript compares the antioxidant capacity and contents of some bioactive compounds with putative medicinal value in Melia azedarach in leaves of plants grown under controlled conditions in well-watered versus drought conditions. The experimental setup is rather simple, but the results are well presented and adequately discussed with the main conclusion that dry conditions may enhance contents of some phytochemicals of medicinal value. The weak point of the study is that it the experimental setup is only compares two treatment groups for an experiment performed only once with ‘weakly significant’ differences only. However, since the experiment was performed under controlled conditions in phytotron I would recommend publishing with minor revisions.

The abstract and introduction is well-written and clear, but for readers unfamiliar with Melia azedarach it would be helpful to also describe which countries (line 63)/regions it is grown ‘in temperate to warm climates’ (line 63)? Is it of economic importance and what production form is being used? And how relevant are projected climate scenarios and dry conditions for these regions?

The experimental conditions are not adequately described and needs to be updated before publishing. The ‘light intensity’ or irradiance seems quite high. Is it PAR-light at 400-700nm? How was it measured, i.e. at plant level? What was the light source? How was 20% field capacity measured?

Furthermore, the authors state the plants were under ‘severe stress’ relating to a relative water content being below 80%. Please elaborate on why 80% is important here, and if there is data available it would be interesting to know if there are differences in growth after 20 days of ‘WW vs. WD’? The authors did measure plant height before treatments (line 184). Were young, old or all seedling leaves collected for analyses?

Suggested specific changes:

Line 65 – change ‘edaphic’ to ‘soil’

Line 70 – change to read ‘…promotes an increased’

Line 75 – change ‘unravel’ to i.e. study, investigate?

Line 81-89 – Abbreviations WD, RWC etc. are introduced before explanations which come later in the text.

Line 106 – Please specify the identities of LC Alkane 1, 2 and 3 referred to in the tables and figures.

The figures and tables appear hastily made, and need improvement.

Figure 1 – There are four graphs assigned letters (A) to (C). Should there be also (D), and the letters are missing to identify which graph they belong?

Figure 2 - The compound names on top and the right side does not align with the lines of squares with colours for all compounds.

Table 1 – Please include the concentration unit in the table heading, and move (mg/mg DW) to be above the column with numerical data values. For the column with lines of fold change, it is not easy to deduce what the numerical values of fold change really is. For instance is Sucrose outside +3.7 fold increase? Also, since all significant difference appear only as ‘weakly’ significant (P<0.5), please include the numerical p-values in the table. P-values may be included in the the ‘fold-change’ column.

Author Response

We would like to acknowledge the valuable comments, remarks and suggestions made by the Reviewers that helped us to improve the quality of the manuscript. We marked in yellow colour the changes made in the manuscript suggested by the Reviewers. In what follows we describe the changes on the manuscript and we answer the Reviewers’ comments.

Reviewer 3

The submitted manuscript compares the antioxidant capacity and contents of some bioactive compounds with putative medicinal value in Melia azedarach in leaves of plants grown under controlled conditions in well-watered versus drought conditions. The experimental setup is rather simple, but the results are well presented and adequately discussed with the main conclusion that dry conditions may enhance contents of some phytochemicals of medicinal value. The weak point of the study is that it the experimental setup is only compares two treatment groups for an experiment performed only once with ‘weakly significant’ differences only. However, since the experiment was performed under controlled conditions in phytotron I would recommend publishing with minor revisions.

Comment: The abstract and introduction is well-written and clear, but for readers unfamiliar with Melia azedarach it would be helpful to also describe which countries (line 63)/regions it is grown ‘in temperate to warm climates’ (line 63)? Is it of economic importance and what production form is being used? And how relevant are projected climate scenarios and dry conditions for these regions?

Response: We followed the comment of the Reviewers, please see pg. 2.

Comment: The experimental conditions are not adequately described and needs to be updated before publishing. The ‘light intensity’ or irradiance seems quite high. Is it PAR-light at 400-700nm? How was it measured, i.e. at plant level? What was the light source? How was 20% field capacity measured?

Response: Yes, is a PAR light (wavelengths between 400 and 700 nm) that emitted an intensity of around 500±20 µmol m-2 s-1. At the plant level the irradiation was around 250±20 µmol m-2 s-1. We measured the light intensity of the lamps in the climate chamber using the external quantum sensor of the of the LI- 6400XT (Portable photosynthesis system from LI-COR Bioscience). We added this information in the section 3.1. pg. 7.

The field water capacity is the amount of water held in soil after the excess water drained away. To measure this amount/quantity of water that is retained in the soil we used a scale. We previously weighed the empty pots, the soil in the respective pot, as well as the water needed to saturate the soil (100%FC) in each pot. Plants were weighted every second day, and the water was replaced until the 100% or 20% (according to the respective treatment). We clear this point in the manuscript “The pots were watered at 100% or 20% field capacity by restoring the quantity of water lost every second day (water lost was measured by weighing the pots with a scale)”. Please see pg. 8.

Comment: Furthermore, the authors state the plants were under ‘severe stress’ relating to a relative water content being below 80%. Please elaborate on why 80% is important here, and if there is data available it would be interesting to know if there are differences in growth after 20 days of ‘WW vs. WD’?

Response: We added more information on the issue of severe stress and relative water content (RWC)(please see pg. 3). The RWC is a measure of the plant water status/dehydration status, giving insights that the water deficit imposed resulted in drought stress for the plants. In this case we can affirm that M. azedarach plants were under drought stress conditions. The RWC values below 80% indicate that the plants are exposed to a severe stress condition, which can cause some physiological impairments (e.g., reduction of photosynthesis, growth…). Unfortunately, we do not have data after the 20 days of water deficit treatment.    

Comment: The authors did measure plant height before treatments (line 184). Were young, old or all seedling leaves collected for analyses?

Response: We used 60 days-old plants (two-month-old) with a height of 13.5±3.2cm in the beginning of the treatment. We used expanded leaves. For the RWC, we selected the leaves from the third top node, while for the metabolomics and antioxidants we collected the rest of the leaves until the last second node from the bottom. We explain this better in this new version of the manuscript (please see pg. 8).

Suggested specific changes:

Comments: Line 65 – change ‘edaphic’ to ‘soil’

Line 70 – change to read ‘…promotes an increased’

Line 75 – change ‘unravel’ to i.e. study, investigate?

Response: We corrected the words. Please see pgs. 2 and 3.

Comment: Line 81-89 – Abbreviations WD, RWC etc. are introduced before explanations which come later in the text.

Response: We explained the meaning of these abbreviations, please see pg.3.

Comment: Line 106 – Please specify the identities of LC Alkane 1, 2 and 3 referred to in the tables and figures.

Response: At the Rt 56.33 min: Tetracontane [(C40H82); m/z 562 (1%)]* that correspond to the LC Alkane 1, at the RT 61.12 min:  Hexatriacontane [(C36H74); m/z 506 (100%) that correspond to the LC Alkane 2, and at the 66.16 min: Dopentacontane [(C52H106); m/z 730 (1%)]*that correspond to the LC Alkane 3 (*the main fragments are, respectively m/z 281 and m/z 365). We added the names of the alkanes in the table and figure (please see pgs. 6 and 7) and in the text (pg. 4).

Comment: The figures and tables appear hastily made, and need improvement.

Response: We did a new version of figure 1 with a new colour legend (pg. 7), and we rearrange Table 1 (pg. 6) adding the P values and the fold change values near the respective bar, and the name of the alkanes. We believe that new version of figure and table is clearer.

Comments: Figure 1 – There are four graphs assigned letters (A) to (C). Should there be also (D), and the letters are missing to identify which graph they belong? Figure 2 - The compound names on top and the right side does not align with the lines of squares with colours for all compounds.

Response: We followed the Reviewer comments. Please see pgs. 4 and 7 (please see also the response to the previous comment).

Comment: Table 1 – Please include the concentration unit in the table heading, and move (mg/mg DW) to be above the column with numerical data values. For the column with lines of fold change, it is not easy to deduce what the numerical values of fold change really is. For instance is Sucrose outside +3.7 fold increase? Also, since all significant difference appear only as ‘weakly’ significant (P<0.5), please include the numerical p-values in the table. P-values may be included in the the ‘fold-change’ column.

Response: We followed the Reviewer comments. In Table 1 (pg. 6) we added the P values and also the fold change values near the respective bar. We believe that new version of figure and table is clearer.

Round 2

Reviewer 2 Report

The authors have significantly improved the manuscript and adequately responded to all my queries. So, I would recommend the manuscript for publication.